# Virtual Screening and Quantum Chemistry Analysis for SARS-CoV-2 RNA-Dependent RNA Polymerase Using the ChEMBL Database: Reproduction of the Remdesivir-RTP and Favipiravir-RTP Binding Modes Obtained from Cryo-EM Experiments with High Binding Affinity

**DOI:** 10.3390/ijms231911009

**Published:** 2022-09-20

**Authors:** Motonori Tsuji

**Affiliations:** Institute of Molecular Function, 2-105-14 Takasu, Misato-shi, Saitama 341-0037, Japan; motonori@molfunction.com

**Keywords:** SARS-CoV-2, COVID-19, RNA-dependent RNA polymerase, virtual screening, quantum chemistry calculation, drug repositioning, ONIOM geometry optimization calculation, frequency analysis, fragment molecular orbital calculation

## Abstract

The novel coronavirus, severe acute respiratory syndrome coronavirus 2 (SARS-CoV-2), was identified as the pathogenic cause of coronavirus disease 2019 (COVID-19). The RNA-dependent RNA polymerase (RdRp) of SARS-CoV-2 is a potential target for the treatment of COVID-19. An RdRp complex:dsRNA structure suitable for docking simulations was prepared using a cryo-electron microscopy (cryo-EM) structure (PDB ID: 7AAP; resolution, 2.60 Å) that was reported recently. Structural refinement was performed using energy calculations. Structure-based virtual screening was performed using the ChEMBL database. Through 1,838,257 screenings, 249 drugs (37 approved, 93 clinical, and 119 preclinical drugs) were predicted to exhibit a high binding affinity for the RdRp complex:dsRNA. Nine nucleoside triphosphate analogs with anti-viral activity were included among these hit drugs, and among them, remdesivir-ribonucleoside triphosphate and favipiravir-ribonucleoside triphosphate adopted a similar docking mode as that observed in the cryo-EM structure. Additional docking simulations for the predicted compounds with high binding affinity for the RdRp complex:dsRNA suggested that 184 bioactive compounds could be anti-SARS-CoV-2 drug candidates. The hit bioactive compounds mainly consisted of a typical noncovalent major groove binder for dsRNA. Three-layer ONIOM (MP2/6-31G:AM1:AMBER) geometry optimization calculations and frequency analyses (MP2/6-31G:AMBER) were performed to estimate the binding free energy of a representative bioactive compound obtained from the docking simulation, and the fragment molecular orbital calculation at the MP2/6-31G level of theory was subsequently performed for analyzing the detailed interactions. The procedure used in this study represents a possible strategy for discovering anti-SARS-CoV-2 drugs from drug libraries that could significantly shorten the clinical development period for drug repositioning.

## 1. Introduction

A novel coronavirus (severe acute respiratory syndrome coronavirus 2 (SARS-CoV-2)) with strong contagious and infective characteristics was identified as the pathogen of coronavirus disease 2019 (COVID-19), which has caused a global pandemic. Several thousands of variants of SARS-CoV-2 exist, and several notable variants that mainly arose from mutations in the receptor-binding domain of the viral spike protein have emerged. Although only a few drugs, such as remdesivir, molnupiravir, baricitinib, and dexamethasone, were approved or authorized for emergency use to treat COVID-19, drugs with greater efficacy and safety are strongly desired. The discovery of potential anti-SARS-CoV-2 drugs from known drug libraries is considered an effective drug repositioning strategy for shortening the clinical development period. Based on this strategy, I have reported anti-SARS-CoV-2 drug candidates through the virtual screening of 1.5 million compounds targeting the main coronavirus protease (M^pro^, also called 3CL^pro^) [1]. Vigorous efforts to identify effective drugs were exerted by researchers using in vitro, in vivo, and in silico strategies from the perspective of drug repositioning and drug repurposing.

SARS-CoV-2 belongs to the betacoronavirus group. One of the best-characterized drug targets among the viral constituents is RNA-dependent RNA polymerase (RdRp), which consists of a catalytic core subunit known as nonstructural protein 12 (nsp12) and the accessory subunits nsp7 and nsp8. Cryo-electron microscopy (cryo-EM) structures of the RdRp complex (nsp12–nsp7–nsp8) with template–primer double-stranded RNA (RdRp complex:dsRNA) were resolved with remdesivir-ribonucleoside triphosphate (RTP) [2,3,4,5] and favipiravir-RTP (Figure 1) [6,7] or without an inhibitor [8]. In the case of remdesivir-RTP, the cryo-EM structure suggested that the covalently incorporated remdesivir-ribonucleoside monophosphate (RMP) could terminate or delay the elongation of the primer RNA with steric repulsion, thus inhibiting transcription and replication [2,4,5]. However, side effects, toxicity, and lower potency limited the use of these drugs in the treatment of COVID-19. Therefore, noncovalent inhibitors with high binding affinity are more suited for the treatment of such viral infections.

In this study, I performed a stepwise structure-based virtual screening using two different docking simulations to discover potential drugs that target the RdRp complex:dsRNA using the ChEMBL database [9], which contains drugs and known bioactive compounds. I expected that the potential drug candidates with anti-SARS-CoV-2 activity obtained from the previous screening, which targeted RdRp (nsp12) largely differed from the results of this study since the previous studies were performed without nsp7, nsp8, and dsRNA [10,11,12,13,14,15,16,17]. Interestingly, nine nucleoside triphosphate analogs (adenosine triphosphate, entecavir triphosphate, favipiravir-RTP, remdesivir-RTP, penciclovir triphosphate, lamivudine triphosphate, GTP, GS-461203, 2-chlorodeoxyadenosine triphosphate), which could function as prodrugs, were obtained as hit compounds with high binding affinity. Remdesivir-RTP and favipiravir-RTP obtained via virtual screening of the ChEMBL database adopted similar docking modes as those observed in these cryo-EM structures, suggesting that the RdRp complex:dsRNA structure might be suitable for discovering potential anti-SARS-CoV-2 drugs. Cryo-EM structures (PDB IDs: 7DOK and 7DOI) of the RdRp complex:dsRNA, bound to penciclovir, became available while this manuscript was under submission. This fact strongly supported the results of this study. Additionally, I performed quantum and molecular mechanics (QM/MM) ONIOM geometry optimization calculations to estimate the binding free energy in the gas phase and, subsequently, fragment molecular orbital (FMO) calculations to analyze the detailed interactions between the RdRp complex:dsRNA and the bioactive compounds. The structural information and interaction analysis of the potential drugs would be useful for improving their pharmacokinetic properties and developing specific anti-SARS-CoV-2 drugs.

## 2. Results and Discussion

### 2.1. Structure-Based Virtual Screenings of the ChEMBL Database Predict Known Anti-SARS-CoV-2 Drugs

In the ChEMBL database, drugs, including approved, clinical, and preclinical drugs, constituted approximately 0.6% of the total number of compounds. The remaining compounds were primarily bioactive compounds, the synthesis of which was, therefore, promising. The advantage of using the ChEMBL database is that it covers all types of drugs from the preclinical stage to approval. I expected that the hit compounds would largely differ from the candidates obtained from the virtual screenings using the focused and targeted libraries [10,11,12,13,14,15,16,17]. Concerning drug repositioning, the ChEMBL database is more suitable for searching for effective known drugs or bioactive compounds when an urgent therapy is necessary and effective drugs are not known.

Appendix A presents the 249 potential drugs that exhibited high binding affinity with the RdRp complex:dsRNA, together with the drug information collected from the ChEMBL web server using the KNIME (Appendix A was sorted according to drug action). The rDock score threshold of ≤−60 kcal·mol^−1^ indicated a relatively high binding affinity with the RdRp complex:dsRNA [1]. Additionally, the rDock score threshold of ≤−60 kcal·mol^−1^ was a reasonable value since the most stable docking mode of remdesivir-RTP, and favipiravir-RTP showed −62.583 kcal·mol^−1^ (AutoDock Vina score = −8.9 kcal·mol^−1^) and −67.292 kcal·mol^−1^ (AutoDock Vina score = −9.1 kcal·mol^−1^), respectively (Appendix A). Based on this energy threshold, I identified 37 approved, 93 clinical, and 119 preclinical drugs from the hit compounds (88,250 distinct compounds with 168,954 docking modes), and the remaining 88,001 were bioactive compounds. For the 249 drugs, the AutoDock Vina scores of the most stable docking modes are also presented in Appendix A. The 249 drugs were largely classified as anti-allergic, anti-bacterial, anti-diabetic, anti-fungal, anti-hypertensive, anti-inflammatory, anti-neoplastic, anti-viral, cardiovascular, gastrointestinal, and neuropsychiatric drugs (Appendix A). Interestingly, the potential drugs included 23 compounds with potential anti-SARS-CoV-2 activity that were identified in a previous study focused on the agents targeting SARS-CoV-2 M^pro^ (Appendix A) [1]. Moreover, nine nucleoside triphosphate analogs (adenosine triphosphate, entecavir triphosphate, favipiravir-RTP, remdesivir-RTP (RDV-TP), penciclovir triphosphate, lamivudine triphosphate, GTP, GS-461203, and 2-chlorodeoxyadenosine triphosphate) with anti-viral activity were included in these hit drugs (Figure 2), among which, remdesivir-RTP and favipiravir-RTP adopted a similar docking mode as that observed in the respective cryo-EM structures (PDB IDs: 7BV2 and 7AAP). The cryo-EM structures (PDB IDs: 7DOI and 7DOK) of the RdRp complex:dsRNA bound to penciclovir became available while this manuscript was under submission. I found that the docking mode of penciclovir was also very similar to that of the experimental cryo-EM structure. Four repurposed drugs (miransertib, prexasertib, zotatifin, and LY-2608204) for SARS-CoV-2 were also obtained in addition to remdesivir-RTP and favipiravir-RTP (Appendix A).

Figure 3 presents the most stable docking modes of remdesivir-RTP (Figure 3A; AutoDock Vina score = −8.9 kcal·mol^−1^) and favipiravir-RTP (Figure 3B,C; AutoDock Vina score = −9.1 kcal·mol^−1^) obtained from the AutoDock Vina docking simulations together with the binding mode of remdesivir-RMP incorporated into the primer RNA (PDB ID: 7BV2 [2]) and favipiravir-RTP (PDB IDs: 7AAP [6] and 7CTT [7]) observed in the cryo-EM structures. The most stable docking mode of penciclovir triphosphate (Figure 3D; AutoDock Vina score = −8.7 kcal·mol^−1^) obtained from the AutoDock Vina docking simulations together with the binding mode of penciclovir monophosphate incorporated into the primer RNA (PDB ID: 7DOI) observed in the cryo-EM structures was also provided in Figure 3D. In this study, I used the RdRp cryo-EM structure (PDB ID: 7AAP) with the catalytically nonproductive favipiravir-RTP conformation [6] because the template–primer double-stranded RNA sequences (template: 3′-AAUUCAAUACUU and primer: 5′-UUAAGUUAU) at the active site of nsp12 were identical to those in the RdRp complex:dsRNA bound with remdesivir-RMP (PDB ID: 7BV2). Conversely, the RdRp cryo-EM structure with the catalytically productive favipiravir-RTP conformation was observed. Figure 3C presents the most stable docking mode of favipiravir-RTP obtained from the docking simulation with the catalytically productive favipiravir-RTP conformation observed in the cryo-EM structure (PDB ID: 7CTT). The triphosphate moiety of favipiravir-RTP of the most stable docking mode adopted a similar conformation as that of the productive conformation.

These results strongly indicate that the refined RdRp complex:dsRNA structure in this study was suitable for screening potential drug candidates from a compound library.

### 2.2. Noncovalent Bioactive Compounds Are Predicted as Potential Inhibitor Candidates for the SARS-CoV-2 RdRp Complex:dsRNA

Appendix A presents the 184 hit compounds obtained using the AutoDock Vina virtual screenings with ≤−12 kcal·mol^−1^ of empirical binding free energy for the RdRp complex:dsRNA. All 184 hit compounds were bioactive compounds registered to the ChEMBL database. These hit compounds were not developed for common targets (for details, see Appendix A), although they could structurally be categorized as large aromatic compounds similar to a typical major or minor groove binder (Figure 4).

The most stable docking modes of the top-scoring eight bioactive compounds are shown in Figure 5. It seems that these compounds mainly interacted with the five ribonucleotides of the 3′-terminal primer RNA (5′-UUAA**GUUAU**-3′), intercalating at the major groove and simultaneously occupying the active site of nsp12, thus stalling the elongation of the primer RNA strand by inhibiting the incoming ribonucleoside triphosphate.

### 2.3. QM/MM ONIOM Geometry Optimization Calculations, Frequency Analyses, and FMO Calculations Are Useful for Improving Pharmacokinetic Properties and Developing Specific Anti-SARS-CoV-2 Drugs

To analyze the detailed interactions with the RdRp complex:dsRNA, three-layer ONIOM geometry optimization calculations in the MP2/6-31G:AM1:AMBER scheme were carried out for the top-scoring bioactive compound, i.e., CHEMBL4246021 (AutoDock Vina score = −13.9 kcal·mol^−1^; Appendix A and Figure 4). Using the ONIOM optimized structure, the binding free energy (*ΔG*^bind^) was calculated at 298.15 K in the gas phase (Table 1). From the single-point frequency analysis for the converged complex (MP2/6-31G:AMBER), isolated compound (MP2/6-31G) and RdRp complex:dsRNA (AMBER), the *ΔG*^bind^_(MP2/6-31G:AMBER)_ was predicted to be −157.799 kcal·mol^−1^. This value was reasonable from the AutoDock Vina score (−13.9 kcal·mol^−1^) [18,19], which is an empirical binding free energy parameterized from the *Ki* values of known protein–inhibitor complexes. 

Subsequently, the FMO calculations using the MP2/6-31G and HF/6-31G methods of the ONIOM optimized system were performed using single-point calculations. I have previously demonstrated that the ONIOM geometry optimization calculations are critical for obtaining reliable interfragment interaction energy (IFIE) values from the FMO single-point calculations [18,19]. Table 2 shows the IFIEs of the components (i.e., nsp12, two nsp8, nsp7, primer RNA, template RNA, metal ions, and pyrophosphate) between an electronically neutral compound, CHEMBL4246021. The IFIE differences between the MP2/6-31G and HF/6-31G methods correspond to the electron correlation energy.

From Table 2, the RdRp complex:dsRNA was strongly stabilized via interacting with CHEMBL4246021 (Table 2; Total). Most of this stability from interacting with an electronically neutral compound, CHEMBL4246021, was attributed to the interactions among two magnesium ions (for MP2/6-31G, IFIEs = −303.211 kcal·mol^−1^ and −165.000 kcal·mol^−1^, respectively, and for HF/6-31G, IFIEs = −169.661 kcal·mol^−1^ and −47.643 kcal·mol^−1^, respectively) at the active site of nsp12 (Figure 6). Zinc ions and pyrophosphate hardly interacted with this bioactive compound. Two nsp8 subunits (nsp8-1 and nsp8-2) and the nsp7 subunit also did not show any significant interactions with this compound. Unexpectedly, nsp12 was destabilized by its interaction with this compound at the HF level of theory; this was mainly attributed to the interaction with the Asp618 residue (IFIE = +39.471 kcal·mol^−1^ at MP2/6-31G and +44.831 kcal·mol^−1^ at HF/6-31G). Asp760 (IFIE = +18.011 kcal·mol^−1^ at MP2/6-31G and +22.144 kcal·mol^−1^ at HF/6-31G) and Asp761 (IFIE = +11.667 kcal·mol^−1^ at MP2/6-31G and +11.815 kcal·mol^−1^ at HF/6-31G) at the active site of nsp12 were slightly destabilized by interacting with this compound. Conversely, Lys545 (IFIE = −20.021 kcal·mol^−1^ at MP2/6-31G and −13.163 kcal·mol^−1^ at HF/6-31G), Lys551 (IFIE = −34.679 kcal·mol^−1^ at MP2/6-31G and −29.306 kcal·mol^−1^ at HF/6-31G), Arg555 (IFIE = −12.969 kcal·mol^−1^ at MP2/6-31G and −6.016 kcal·mol^−1^ at HF/6-31G), and Lys798 (IFIE = −15.920 kcal·mol^−1^ at MP2/6-31G and −12.491 kcal·mol^−1^ at HF/6-31G) at the active site of nsp12 were stabilized by interacting with this compound. The ribonucleotides of template–primer double-stranded RNA, in particular G16 (IFIE = −11.488 kcal·mol^−1^ at MP2/6-31G and −9.768 kcal·mol^−1^ at HF/6-31G), U18 (IFIE = −15.909 kcal·mol^−1^ at MP2/6-31G and −9.922 kcal·mol^−1^ at HF/6-31G), and A19 (IFIE = −14.164 kcal·mol^−1^ at MP2/6-31G and −6.983 kcal·mol^−1^ at HF/6-31G) of primer RNA in the major groove, were stabilized by interacting with this compound, as expected.

The detailed results of the FMO analysis for CHEMBL4246021 are summarized in Figure 7. Two magnesium ions and the charged amino acid residues at the active site of RdRp (nsp12) participated in the intercalation between the primer RNA and this compound.

Although the structural features of the hit bioactive compounds depend on the primer and template RNA sequences, these results suggest that these bioactive compounds may function through the same underlying mechanism. It was suggested that the intercalation of these bioactive compounds would inhibit transcription and replication. Bioactive compounds with high binding affinity for the SARS-CoV-2 RdRp complex:dsRNA could be used as a basis for improving pharmacokinetic properties and developing specific anti-SARS-CoV-2 drugs.

## 3. Materials and Methods

### 3.1. Structural Refinement of the RdRp Complex:dsRNA

I prepared an RdRp complex:dsRNA structure suitable for docking simulations using a cryo-EM structure (PDB ID: 7AAP [6]; resolution, 2.60 Å). The structural refinement was performed using the Homology Modeling Professional for HyperChem (HMHC) software I developed [20,21,22], and energy calculations were performed under the AMBER99 force field using the following parameters: root-mean-square gradient, 1.0 kcal·mol^−1^·Å^−1^; algorithm, Polak–Ribière; cut-off, none; 1–4 van der Waals scale factor, 0.5; 1–4 electrostatic scale factor, 0.833; dielectric scale factor, 1.0; and distance-dependent dielectric condition. Structural refinement was conducted in the presence of favipiravir-RTP. After adding hydrogen atoms automatically, the missing residues (Thr856–Asp920) of nsp12 were modeled using the HMHC software, and I assigned the Mulliken atomic charges of favipiravir-RTP and pyrophosphate using the single-point calculations of the semiempirical MNDO/d method. The Mulliken atomic charges obtained via the MNDO/d calculation displayed an empirically good correlation with the AMBER charges [23]. In addition, AMBER99 atom types were assigned. The N- and C-termini of the RdRp complex (nsp12, nsp7, and two nsp8 subunits) were treated as zwitterions, the aspartic and glutamic acid residues were treated as anions, and the lysine, arginine, and histidine residues were treated as cations under physiological conditions. The 5′- and 3′-termini of the dsRNA were treated as OH and PO_3_H, respectively. Next, Mg^2+^ and Zn^2+^ ions included in the cryo-EM structure were assigned +2 charges. Subsequently, partial optimization with Belly calculations for all components, excluding heavy atoms, was performed, and distance restraint conditions (7.0 kcal·mol^−1^·Å^−2^) were applied to all heavy atoms of the aforementioned structure. Next, geometry optimization calculations were performed. The resulting structure was subjected to low-temperature molecular dynamics annealing (starting temperature, 0 K; heat time, 30 ps; simulation temperature, 300 K; run time, 100 ps; final temperature, 0 K; cooling time, 30 ps; step size, 0.001 ps; and temperature step, 0.01 K). Finally, all distance restraint conditions were removed, and the structure was further optimized to obtain the final structure. The precision of the final structure was confirmed using the Ramachandran plot program of the HMHC.

### 3.2. Preparation of the 3D Structure Database from the ChEMBL Database

The planar structures of the ChEMBL database (ChEMBL28; 2,066,376 distinct compounds, including more than 13,000 drugs) [9] were downloaded from the ChEMBL website in the SDF file format. MayaChemTools (2021) [24] was used to remove the counterions and inorganic compounds from the database. Then, 3D structures were obtained using Balloon version 1.6.9 [25] under an MMFF94 force field. The resulting 3D structure database was treated using Babel version 2.4.1 [26]. The compounds’ protonation state was prepared under physiological conditions (pH = 7.4) and filtered by molecular weight (MW ≥100 and ≤700) to reduce the database to a more drug-like library. In total, 1,838,257 compounds were used for the subsequent virtual screenings.

### 3.3. Stepwise Structure-Based Virtual Screenings

Structure-based virtual screenings were performed using rDock (2013) [27] and AutoDock Vina version 1.1.2 [28]. Both interfaces are available in the Docking Study with HyperChem (DSHC) software I developed [20,29], and the resulting docking modes filtered by the rDock score threshold were more precisely simulated using AutoDock Vina.

Prior to the docking simulations, favipiravir-RTP was removed from the RdRp complex:dsRNA system. Then, docking simulations for the 3D structure database (1,838,257 compounds) were performed using the relatively reliable, high-speed docking simulation program rDock under the default condition. The cavity condition for the rDock docking simulations was prepared using favipiravir-RTP (the reference ligand method) under a default condition. The docking simulations outputted three docking modes per trial compound in the SDF file format. Consequently, 5,507,523 docking modes were obtained (rDock could not assess or support 2416 out of 1,838,257 compounds). These docking modes were filtered by the rDock score threshold of ≤−60 kcal·mol^−1^ to obtain 88,250 distinct compounds (168,954 docking modes) in the SDF file format. The ChEMBL IDs of these distinct compounds were subjected to KNIME version 4.1.2 [30] to collect the compound information from the ChEMBL web server. Some information was manually collected from the Kyoto Encyclopedia of Genes and Genomes database [31].

From the 168,954 docking modes obtained via the virtual screenings, the 88,250 distinct hit compounds had two docking modes on average. The hit compounds, including the 249 drugs I found, were more precisely investigated using the AutoDock Vina docking simulations with these docking modes as the initial structures. Subsequently, the resulting 168,954 docking modes were separated and converted into individual PDBQT files using DSHC (some compounds, including unsupported metal atoms, such as boron and selenium, were rejected at this stage). Then, more precise docking simulations were performed using the AutoDock Vina In Silico Screenings Interface software integrated into DSHC. The aforementioned RdRp complex:dsRNA system in the PDB file format was also converted to a PDBQT file using DSHC. A configuration file with cavity information (the value of the grid box was set to center_x = 100.646 Å, center_y = 96.712 Å, center_z = 112.158 Å with size_x = 22.254 Å, size_y = 24.537 Å, and size_z = 22.606 Å) was prepared using DSHC, and other docking conditions were set to the default values (the top nine docking modes per trial compound were maximally outputted). Docking simulations with AutoDock Vina produced 1,508,525 docking modes, which were filtered by the AutoDock Vina score (empirical binding free energy) threshold of −12 kcal·mol^−1^. Because the AutoDock Vina score reflects empirical binding free energy, I expected that a score of −11 kcal·mol^−1^ would theoretically represent a subnanomolar order of binding affinity with the RdRp complex:dsRNA. When the threshold for screening was set to less than this value, I obtained 2259 distinct compounds (6560 docking modes) as hit compounds. To more realistically concentrate the number of hit compounds, I set the threshold value to ≤−12 kcal·mol^−1^. Consequently, I obtained 184 distinct compounds (353 total docking modes). The ChEMBL IDs of these distinct compounds were subjected to KNIME to collect the compound information from the ChEMBL web server.

### 3.4. ONIOM Geometry Optimization and Frequency Analysis Calculations

For the top-scoring (i.e., the most stable in energy) bioactive compound, i.e., CHEMBL4246021, a three-layer ONIOM calculation was performed using the Gaussian16 program [32]. Prior to the ONIOM calculation [33], the hydrogen atoms of the docked compound of the complex were prepared, and MNDO/d Mulliken atomic charges were assigned. Subsequently, the compound structure was further optimized using the AMBER99 force field. A Gaussian job file for the complex structure was automatically prepared using the ONIOM Interface for Receptor software integrated into the HMHC [20]. Three-layer ONIOM geometry optimization calculations (MP2/6-31G:AM1:AMBER) were performed; the CHEMBL4246021 was defined as the high layer (MP2/6-31G), the amino acid residues, rebonucleotides, and metal ions around 6 Å from the CHEMBL4246021 were defined as the medium layer (AM1), and the remaining structures were defined as the low layer (AMBER). In this calculation, the structures of the high and medium layers were fully optimized, whereas only hydrogen atom positions were optimized for the low layer. The binding free energies (*ΔG*^bind^) at 298.15 K in the gas phase were obtained from the single-point frequency analysis for the converged complex (using two-layer ONIOM calculations; MP2/6-31G:AMBER), the isolated CHEMBL4246021 (MP2/6-31G), and the remaining structure (AMBER) [18,19].

### 3.5. FMO Calculations

For the ONIOM converged structure, the FMO calculation was performed at the MP2 level of theory using the ABINIT-MP program [34,35]. The job file was manually prepared as follows: the CHEMBL4246021, pyrophosphate, magnesium ions, and zinc ions were treated as single fragments, each amino acid residue of the RdRp complex was treated as a single fragment, and each ribonucleotide of dsRNA was treated as single fragments. The interfragment interaction energies (IFIEs) were obtained using single-point calculations in the MP2/6-31G and HF/6-31G methods. The BioStation Viewer software [36] was used to analyze the interactions among these fragments (a total of 1163 fragments).

## 4. Conclusions

This study was performed to rapidly identify potential anti-SARS-CoV-2 drug candidates using drug repositioning and to predict drugs specifically developed for the treatment of COVID-19. Noncovalent major and minor groove binders with high binding affinity for the SARS-CoV-2 RdRp complex:dsRNA could be used as a basis for developing specific anti-SARS-CoV-2 drugs. The combinations of structure-based docking simulations, as well as subsequent QM/MM ONIOM geometry optimizations and FMO calculations, are valuable for analyzing the underlying mechanism of hit bioactive compounds. Determination of the effect of the potential anti-SARS-CoV-2 drugs obtained in this study is in progress [37].

In conclusion, the method described in this study proved useful for developing specific anti-SARS-CoV-2 drugs. The results highlighted the potential utility of the identification strategy for quickly repurposing drugs for the treatment of COVID-19.

## Figures and Tables

**Figure 1 ijms-23-11009-f001:**
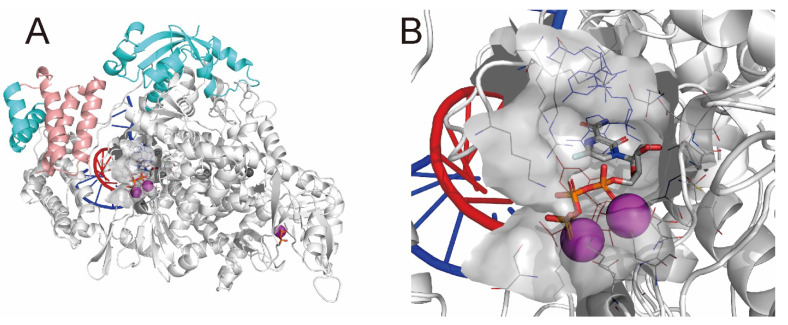
Refined cryo-EM structure (PDB ID: 7AAP) of the SARS-CoV-2 RdRp complex:dsRNA bound with favipiravir-RTP. (**A**) The whole structure and (**B**) the enlarged structure of the active site. RdRp (nsp12), two nsp8 subunits, and one nsp7 subunit are presented as white, cyan, and pink ribbons, respectively. Primer and template RNAs are presented as red and blue coils, respectively. Favipiravir-RTP is presented in CPK color using tubes. Pyrophosphate is presented using tubes. Magnesium and zinc ions are presented as magenta and dark gray spheres, respectively. Residues located 5 Å from the favipiravir-RTP are presented using lines without hydrogen atoms. The van der Waals surfaces of the active sites are presented in gray.

**Figure 2 ijms-23-11009-f002:**
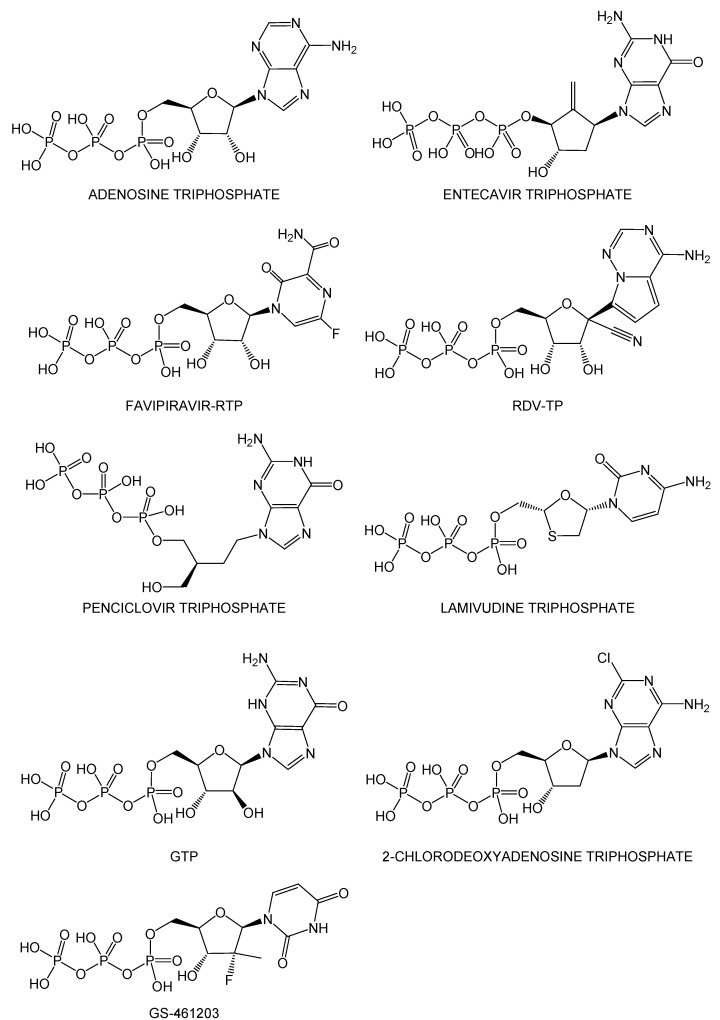
Planar structures of nine nucleoside triphosphate analogs with anti-viral activity obtained from rDock docking simulations.

**Figure 3 ijms-23-11009-f003:**
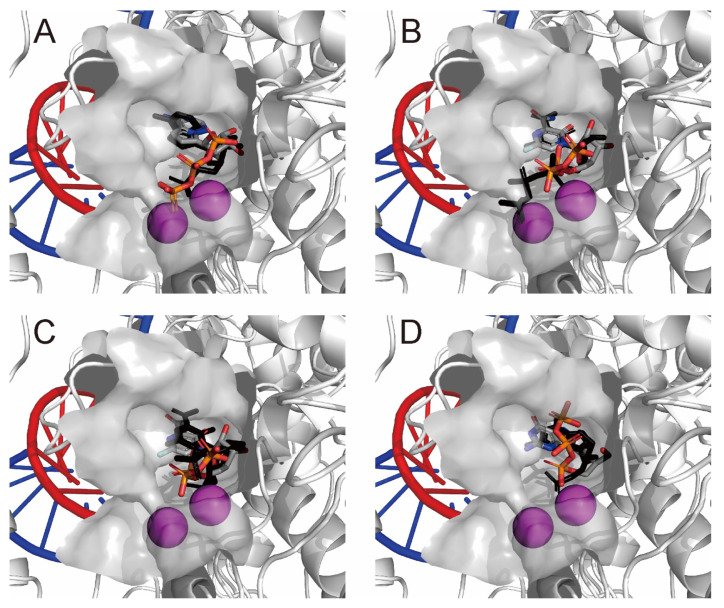
The most stable docking mode obtained from the AutoDock Vina docking simulations using the docking mode from the rDock virtual screenings of the ChEMBL database. (**A**) Binding mode of remdesivir-RTP. (**B**,**C**) Binding mode of favipiravir-RTP. (**D**) Binding mode of penciclovir triphosphate. RNA-dependent RNA polymerase (nsp12) is presented as a white ribbon. Primer and template RNAs are presented as red and blue coils, respectively. The compound is presented in CPK color using tubes. Magnesium ions are presented as magenta spheres. The experimental drugs remdesivir-RMP (**A**, PDB ID: 7BV2), favipiravir-RTP (**B**, PDB ID: 7AAP; **C**, PDB ID: 7CTT), and penciclovir monophosphate (**D**, PDB ID: 7DOI) are presented as black tubes. The van der Waals surface of the active site is presented in gray. Hydrogen atoms are not shown.

**Figure 4 ijms-23-11009-f004:**
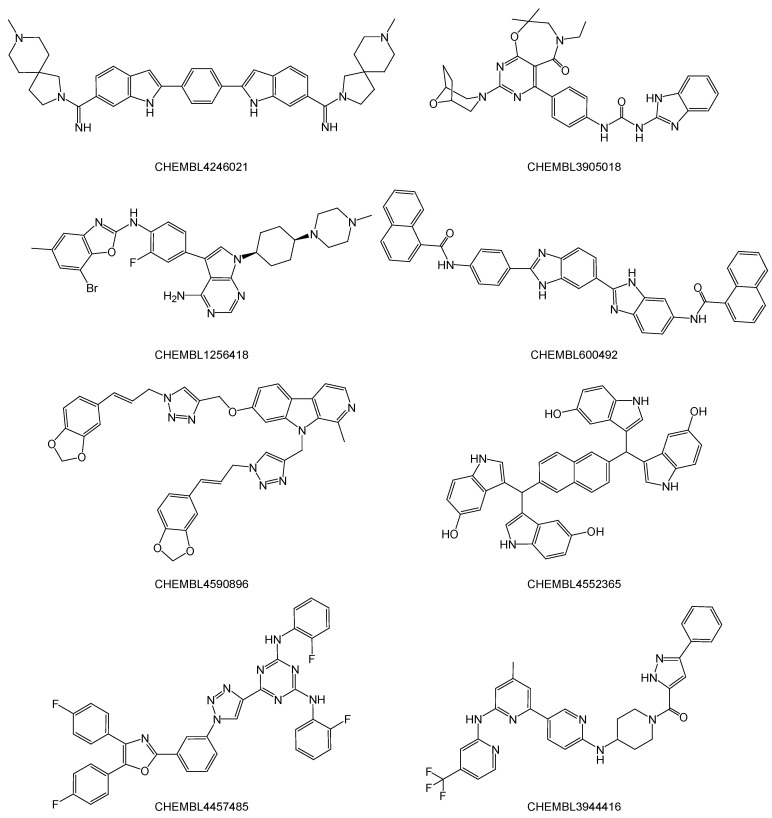
Planar structures of the top-scoring eight (≤−13 kcal·mol^−1^) bioactive compounds obtained from the AutoDock Vina docking simulations.

**Figure 5 ijms-23-11009-f005:**
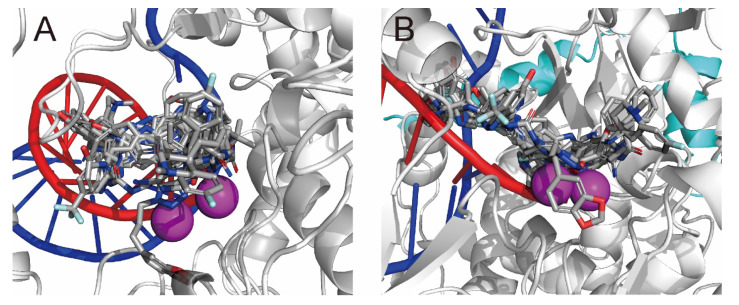
Superposition of the most stable docking modes of the top-scoring eight bioactive compounds obtained from the AutoDock Vina docking simulations. (**A**) Front view and (**B**) side view of the active site. The RdRp (nsp12) and nsp8 subunits are presented as white and cyan ribbons, respectively. The primer and template RNAs are presented as red and blue coils, respectively. The bioactive compounds are presented in CPK color using tubes. The magnesium ions are presented as magenta spheres.

**Figure 6 ijms-23-11009-f006:**
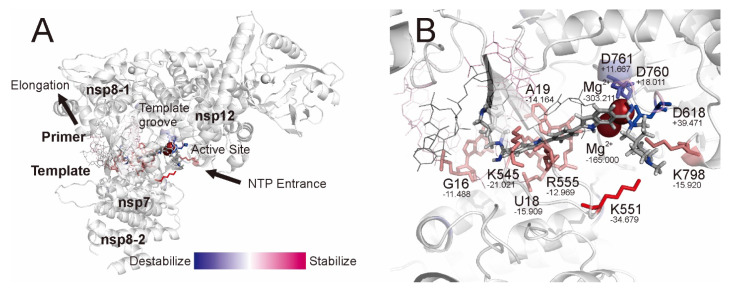
The IFIEs at the MP2/6-31G level of theory mapped on the three-layer ONIOM optimized structure of the RdRp complex:dsRNA with CHEMBL4246021. (**A**) The whole structure and (**B**) enlarged structure. RdRp (nsp12), two nsp8 subunits, and one nsp7 subunit are presented using ribbons. The primer and template RNAs are indicated using lines. CHEMBL4246021 is presented in CPK color using tubes. The magnesium and zinc ions are shown with spheres. Pyrophosphate is presented using tubes. The amino acid residues and ribonucleotides interacting with CHEMBL4246021 are presented as colored tubes. All hydrogen atoms, excluding CHEMBL4246021, are not shown. The numerical values under the residue label show the IFIE values (kcal·mol^−1^) between CHEMBL4246021. NTP: nucleotide triphosphate.

**Figure 7 ijms-23-11009-f007:**
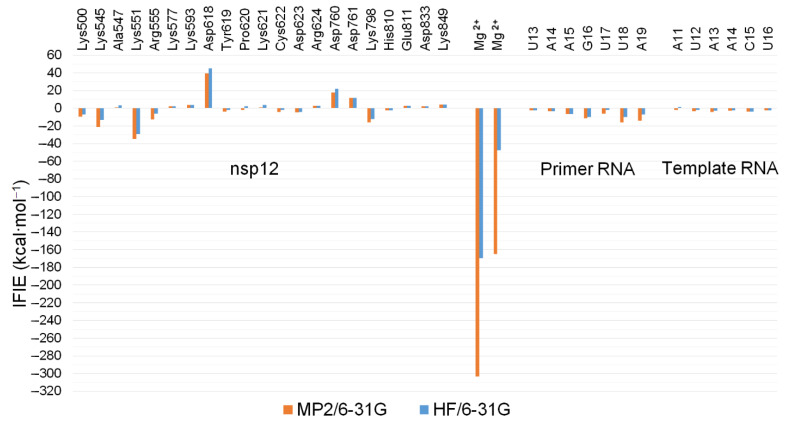
The IFIEs between the components of the RdRp complex:dsRNA and CHEMBL4246021 at the MP2/6-31G and HF/6-31G methods. Only the amino acid residues, ribonucleotides, and metal ions with interaction energies of more than +2 kcal·mol^−1^ or less than −2 kcal·mol^−1^ are shown.

**Table 1 ijms-23-11009-t001:** Binding free energy between the RdRp complex:dsRNA and CHEMBL4246021 at 298.15 K in the gas phase.

Component	Method	*G* (Hartree) at 298.15 K and 1 atm	*ΔG*^bind^ (kcal·mol^−1^) at 298.15 K and 1 atm
RdRp Complex:dsRNA−CHEMBL4246021	MP2/6-31G:AMBER	−2017.589	−157.799
RdRp Complex:dsRNA	AMBER	40.782	
CHEMBL4246021	MP2/6-31G	−2058.119	

**Table 2 ijms-23-11009-t002:** IFIEs (kcal·mol^−1^) between the components of the RdRp complex:dsRNA and CHEMBL4246021 at the MP2/6-31G and HF/6-31G methods.

Method	nsp12	nsp8-1 (79–190)	nsp8-2 (85–110)	nsp7	Primer RNA	Template RNA	Metal Ions	Pyrophosphate	Total
MP2/6-31G	−20.859	2.294	0.267	0.223	−60.033	−21.161	−468.308	0.811	−566.766
HF/6-31G	29.672	2.294	0.267	0.223	−40.582	−12.484	−217.402	0.811	−237.201

## Data Availability

The ChEMBL28 dataset is available from https://www.ebi.ac.uk/chembl/. My developed HMHC (with the Gaussian Interface and ONIOM Interface) and DSHC (with the AutoDock Vina In Silico Screenings Interface and rDock Interface) software, which run on HyperChem (http://hypercubeusa.com/) software, are available from Institute of Molecular Function (https://www.molfunction.com/). MayaChemTools (http://www.mayachemtools.org/), Balloon (http://users.abo.fi/mivainio/balloon/), and Babel (http://openbabel.org/wiki/Main_Page/) programs for preparing the 3D structure database are available from the described websites, respectively. rDock (http://rdock.sourceforge.net/) and AutoDock Vina (https://vina.scripps.edu/) programs for docking small molecules are available from the respective websites. KNIME software is available from https://www.knime.com/. The Gaussian16 program is available from http://gaussian.com/. The ABINIT-MP program and BioStation Viewer software are available from http://www.ciss.iis.u-tokyo.ac.jp/riss/english/. The input (job) files, summarized output data, and other relevant datasets are available in the Appendix A. All data prepared in this study are available from the author.

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
