# Peer review of "Virtual Screening and Quantum Chemistry Analysis for SARS-CoV-2 RNA-Dependent RNA Polymerase Using the ChEMBL Database: Reproduction of the Remdesivir-RTP and Favipiravir-RTP Binding Modes Obtained from Cryo-EM Experiments with High Binding Affinity"

_ijms, 2022, doi:10.3390/ijms231911009_

Round 1
Reviewer 1 Report
The author present an interesting work combining computational chemistry methods to search for possible binders of RdRP of COVID-19 virus. I have few observations and comment to make.
What is the significance of -60 kcal/mol in the rDock score threshold? May be a better threshold is the docking scores of the bound complexes with cryo-EM structures.
Readers may get a wrong idea that the cryo-EM structure of the bound complexes are prepared and reported here, based on the Abstract. Please clarify that these already published structures are used for in silico studies.
Did all the targets bind to same binding site? Did the author check for additional binding sites for the compounds? Is it a targeted docking or a blind docking?
As the author obtained a significant number of molecules that can bind to the target protein, what are the most promising pharmacophore model proposed?
The author need to incorporate solvent effect (at least through single point calculation if the system size is prohibitively large) to account for the biological environment in the ONIOM calculation.
lines 304-307: Binding energy from ONIOM calculations and docking energy of AutodockVina is no way comparable to each other. They are first of completely different magnitude and are obtained from different underlying principles.
Is there Mg ion - Ligand interaction present in the complexes? If yes, could you please highlight them, may be as an inset of Figure 6.
Reviewer 2 Report
The paper by Motonori Tsuji presents a thorough virtual screening and docking investigation on compounds from the ChEMBL database on the RdRp:RNA complex, identifying several hits. Among these, small molecules which were subsequently co-crystallized with the receptor:RNA pair show similar docking poses to those in crystal structures, proving that the designed system is suitable for positive hit identification and correct binding pose prediction.
The paper is very easy to follow, scientifically sound and interesting, and the methods used can be adopted for any kind of system in order to contribute to drug discovery efforts in a plethora of systems.
I could not find any methodological errors in the paper, nor did I find any typos.
The only thing I would have liked more clear is the dimension of the gridbox used in AutoDock Vina. Could you provide this information?
In addition, I would change 'developed' in line 377 with 'identifying', as this tool did not aid in the development, but in the identification through virtual screening and docking, of the hit compounds.
Congratulations to the author! Amazingly well built methodology and powerful discussion section.
Round 2
Reviewer 1 Report
This reviewer thank the author for attending to the comments.